# Research

behaviour, evolution

sexual selection, aggression, experimental evolution, *Drosophila melanogaster*, sexual conflict, sex ratio

**Author for correspondence:**
Eleanor Bath
e-mail: eleanor.bath@zoo.ox.ac.uk

†Joint first authors.

# Sex ratio and the evolution of aggression in fruit flies

Eleanor Bath[1,†], Danielle Edmunds[1,†], Jessica Norman[1], Charlotte Atkins[1], Lucy Harper[1], Wayne G. Rostant[2], Tracey Chapman[2], Stuart Wigby[1,3] and Jennifer C. Perry[1,2]

[1]Department of Zoology, University of Oxford, 11a Mansfield Road, Oxford OX1 3SZ, UK
[2]School of Biological Sciences, University of East Anglia, Norwich NR4 7TJ, UK
[3]Department of Evolution, Ecology, and Behaviour, Institute of Infection, Veterinary and Ecological Sciences, University of Liverpool, Liverpool, UK

EB, 0000-0002-2988-8061; DE, 0000-0002-8915-1352; TC, 0000-0002-2401-8120; SW, 0000-0002-2260-2948; JCP, 0000-0002-8449-2764

Aggressive behaviours are among the most striking displayed by animals, and aggression strongly impacts fitness in many species. Aggression varies plastically in response to the social environment, but we lack direct tests of how aggression evolves in response to intra-sexual competition. We investigated how aggression in both sexes evolves in response to the competitive environment, using populations of *Drosophila melanogaster* that we experimentally evolved under female-biased, equal, and male-biased sex ratios. We found that after evolution in a female-biased environment—with less male competition for mates—males fought less often on food patches, although the total frequency and duration of aggressive behaviour did not change. In females, evolution in a female-biased environment—where female competition for resources is higher—resulted in more frequent aggressive interactions among mated females, along with a greater increase in post-mating aggression. These changes in female aggression could not be attributed solely to evolution either in females or in male stimulation of female aggression, suggesting that coevolved interactions between the sexes determine female post-mating aggression. We found evidence consistent with a positive genetic correlation for aggression between males and females, suggesting a shared genetic basis. This study demonstrates the experimental evolution of a behaviour strongly linked to fitness, and the potential for the social environment to shape the evolution of contest behaviours.

## 1. Introduction

Aggressive contests occur in males and females across diverse animal taxa [1]. The nature of aggressive contests often differs between the sexes: males largely compete for reproductive opportunities and females largely for reproductive resources [2]. Because aggression significantly impacts fitness in both sexes [3–5], aggressive contests form an important part of reproductive competition [6–8]. Hence, the intensity of reproductive competition in a population should determine the strength of sexual and social selection on aggressive behaviours [2,9,10].

More intense reproductive competition is predicted to lead to heightened aggression [11]. This prediction has received empirical support. Comparative studies of chernetid false scorpions and dung beetles have found that the presence and size of male weapons is positively correlated with population density and degree of male bias in the sex ratio across species [12,13]. Behavioural studies have reported increased aggression in the sex in excess within populations in fish [14,15]. However, comparative studies cannot eliminate the possibility that variation in aggression is due to other factors that covary

with the intensity of competition, such as conspecific density or resource distribution [16]. Likewise, behavioural studies do not show how the competitive environment shapes diversity in aggression across groups. Hence, direct tests of how aggression evolves in response to the intensity of competition are lacking.

An additional challenge to studying adaptive variation in aggression is that male and female aggression might be constrained by their shared genome, preventing either or both sexes from reaching their optimum [17]. Indeed, intra-sexual aggression has sometimes been considered a predominantly male trait, with female aggression assumed to arise as a by-product of an intersex genetic correlation ([4] and references therein). Recently, female–female aggression has gained attention as an adaptive strategy for maximizing access to resources required for reproduction [8,18], leading to improved reproductive success or offspring survival [19–21]. However, we currently lack data on the independence of the evolution of aggression in each sex.

Beyond constraints through the shared genome, female aggression might also depart from the female optimum if female behaviour is subject to manipulation by males [22]. In polygynous mating systems, the optimal level of female–female aggression will be higher for males than for females whenever female aggression confers immediate reproductive benefits that both mating partners experience, but incurs longer-term costs to females in lifetime reproduction. Mating offers males an opportunity to influence female behaviour through ejaculate transfer, and ejaculate-stimulated changes in female behaviour are well-documented [23]. In several species, shifts in female aggression are associated with mating [20,24,25]. Overall, because female aggression has been under-researched relative to male aggression, key facets of the evolution of female aggression, including sexual conflict, the intersex genetic correlation, and responses to intra-sexual competition, are not yet fully understood.

Here, we used experimental evolution to ask how male and female aggression evolve in response to the intensity of intra-sexual competition. We exposed replicate populations of fruit flies, *Drosophila melanogaster*, to different competitive environments for greater than 75 generations via manipulation of the population sex ratio, a common proxy for the intensity of competition [11,26,27]. Aggression is heritable in *D. melanogaster* and can evolve rapidly under laboratory conditions [28]. Both sexes engage in contests over food patches. For females, food patches provide nutrition required for egg production [29]. For males, which display limited adult feeding [30], food patches predominantly provide access to mates [6,7,31,32]. Both sexes display aggressive behaviours including fencing, male lunging and female head-butting [7,33]. Mating increases female aggression [33,34] due to the effects of sperm and seminal fluid proteins received at mating [35]. Therefore, evolved differences in female aggression could represent a response to evolved differences in male stimulation of aggression—mediated by sexual conflict—as well as the direct evolution of female behaviour.

We addressed the following questions: does the evolutionary sex ratio drive the evolution of male and female aggression? Does the evolutionary sex ratio affect the post-mating increase in female aggression? Is there evidence for a genetic correlation between male and female aggression? We predicted, first, that males and females evolving in a population biased towards their sex would display heightened aggression. Second, if increased aggression after mating is adaptive for females, then we expected a greater increase in aggression after mating in females from female-biased populations. Third, if female aggression responds to the sex ratio through female adaptation, then we expected that sex ratio effects would occur when experimentally evolved females mated with males from stock populations, whereas if female aggression responds to the sex ratio through male adaptation to the sex ratio, then we expected that experimentally evolved males would induce altered aggression in female mates from stock populations. Finally, if the sexes share a genetic basis for aggression, then we expected congruent changes in aggression across populations.

## 2. Methods

### (a) Overview

We conducted two experiments. First, we measured intra-sexual aggression in virgin females, mated females, and mated males that had evolved under male-biased, equal and female-biased evolutionary sex ratios (Experiment 1—'coevolved'). In this experiment, all mated individuals mated with partners from the same replicate population. We tested both virgin and mated females because females show a distinct increase in aggression post-mating [33,35], but tested only mated males because, to our knowledge, male aggression does not change with mating (though there is some evidence for mate guarding [36]). We then conducted a second, two-stage experiment to test whether differences in female aggression among sex ratio treatments arise from the evolution of female aggression itself or of male stimulation of female aggression. To do this, we mated experimentally evolved females with stock males (Experiment 2—'evolved female'), and stock females with experimentally evolved males (Experiment 2—'evolved male'), and measured female aggression before and after mating. Stock individuals were derived from the same wild-type Dahomey background from which experimentally evolved populations were generated.

Experimentally evolved flies were maintained in three independent replicate populations per sex ratio (see electronic supplementary material and [37] for details). We assayed behaviour after 78 generations for the Experiment 1 and 92 generations for Experiment 2. Fly husbandry and experiments were conducted at 25°C on a 12 : 12 h light : dark cycle with uncontrolled humidity.

### (b) Generation of experimental flies

We collected eggs from each of the nine replicate populations and the stock population and raised larvae at a standardized density on standard laboratory medium [38].

At eclosion (day 1), we collected virgin flies under ice anaesthesia. Flies used in aggression trials were housed singly. Males that were used as mates only (in Experiment 2) were housed in pairs. We randomly assigned females to the virgin or the mated treatment. Females assigned to the virgin treatment were housed singly and transferred to new vials on day 3 after eclosion (to mirror how mated females were handled). On day 3, we transferred pairs of males and females (those assigned to the mating treatment) from the same replicate population into fresh vials, recorded mating latency and duration, and separated pairs into individual vials when copulation ended. We discarded pairs that did not mate within 3 h.

### (c) Aggression trials

On day 4, we placed all flies singly into food deprivation vials containing only damp cotton wool for 2 h to increase aggressive

motivation. We randomly assigned flies to a same-sex dyad, with both flies in the dyad coming from the same replicate population and mating status ($N = 10$–$29$ per population; electronic supplementary material, tables S2–S4) to standardize the difference between competitors within contests and to expose individuals to the type of competitor encountered in their recent evolutionary history. We transferred dyads into observation chambers (20 mm diameter, 5 mm depth) containing a central food cup (5 mm diameter, standard laboratory medium and live yeast paste). We randomly assigned dyads a trial time between 2 and 6 h Zeitgeber time and allowed 5 min acclimatization before recording aggression trials of 15 min (Toshiba Camileo X400 cameras). We observed each dyad once and discarded flies after trials.

## (d) Behavioural data extraction

All videos were scored by observers blind to treatment using JWatcher v. 1.0 (Macquarie University & UCLA) and BORIS v. 7.7.3 [39]. We recorded aggressive behaviours as described in electronic supplementary material, table S1. To avoid pseudo-replication, the dyad was taken as the unit of replication, with behaviour measures summed for the two individuals. Lunging, chasing and tussling (in males) and headbutts (in females) represent high-intensity aggression and fencing in both sexes represents low-intensity aggression [32]. We calculated a male high-intensity aggression score by summing the amount of time each dyad spent lunging, chasing and tussling. Because food patches can represent breeding territories for males [16,40], and attractive nutritional resources for females [33,35], we calculated food patch occupancy as the average duration the two flies in a dyad spent on the food patch so that we could assess the relationship between aggression and patch occupancy. We recorded the sum of the duration the two flies in a dyad spent walking to test for locomotor differences that might influence aggression. For females, all videos were scored for headbutts as the main high-intensity aggressive behaviour. A subset was also scored for female fencing so that we could assess whether differences extended to low-intensity aggression.

## (e) Statistical analyses

Statistical analyses were conducted in R v. 3.6.2 (2019-12-12), using packages 'MASS' [41], 'emmeans' [42], 'lme4' [43], 'survminer' [44] and 'coxme' [45]. We identified outliers by inspection of boxplots or, where data were non-normally distributed, adjusted boxplots [46]. We replaced points outside 1.5* the interquartile range with the value of the lower or upper 1.5* interquartile range (i.e. winsorization [47]).

For all experiments, we ran linear mixed effects models (LMMs; lme4 lmer() function) to test the influence of evolutionary sex ratio on the number of lunges (in males) or headbutts (in females), fencing duration, intense male aggression duration, locomotion duration and food patch occupancy. We ran binomial general linear mixed effect models (GLMMs) to test the influence of evolutionary sex ratio on the proportion of male total aggression (fencing, chasing, lunging and tussling) or female headbutting performed on the food patch. For models of female behaviour in Experiment 1—'coevolved' and Experiment 2—'evolved female', we included evolutionary sex ratio, mating status, their interaction and observer as fixed factors. For models of male behaviour in the Experiment 1—'coevolved', we included evolutionary sex ratio as a fixed factor (a single observer extracted male data). All models included replicate population and day as random factors and Zeitgeber time as a covariate, and models of female behaviour in Experiment 1—'coevolved' and Experiment 2—'evolved female' also included the interaction between replicate population and mating status as a random effect. For Experiment 2—'evolved male', we had a single virgin female treatment and three mated female treatments (i.e. stock females

mated to males from each sex ratio). We first assessed the effect of mating on aggression and food occupancy in an LMM with mating status as a fixed factor. For mated females, we then ran a model including evolutionary sex ratio as a fixed factor. Both models included replicate population and day as random factors and Zeitgeber time as a covariate. We found no influence of evolutionary sex ratio on mating latency or duration (electronic supplementary material, table S5), so we did not include mating behaviour as a covariate in any models.

We examined model fit by inspection of diagnostic plots, and where necessary, applied transformations. We analysed LMMs with Wald $F$-tests with Kenward–Roger degrees of freedom [48] (type III for models with significant interactions, type II for models without significant interactions) and analysed binomial GLMMs with Wald $\chi^2$-tests. In female models, when we found a significant interaction between sex ratio and mating status, we re-ran models separately for virgin and mated females to explore sex ratio effects within each group. When sex ratio was significant, we explored the effect using post-hoc Tukey tests. For females, we compared the magnitude of the post-mating changes in behaviours among sex ratios using post-hoc effect size tests.

When we found an effect of evolutionary sex ratio on food patch occupancy, we investigated the relationship between aggression and food patch occupancy. We used binomial general linear mixed models as described above to test whether the individual that performs the greatest proportion of total aggression (in males) or headbutts (in females) within a dyad also spends the highest proportion of time on the food patch, and whether this relationship was influenced by evolutionary sex ratio. Individuals that performed equal aggression (16 male dyads, 24 female dyads) were excluded from this analysis. Full model output for all LMMs is included in electronic supplementary material.

To explore whether the evolution of sex-specific aggression might be constrained by a shared genetic basis between the sexes, we assessed the correlation between the aggressive behaviour of males and females that evolved in the same replicate population, using data from Experiment 1—'coevolved'. A positive correlation might arise from a shared genetic basis, from similar effects of the time and day of behavioural observations in both sexes, or from congruent evolution in response to the evolutionary sex ratio. To control for the influence of time and day (and observer, for female data for which multiple observers were involved) on variation in aggression among vials, we ran linear models of lunging, headbutting and fencing against time and day (and observer, for female data), and used model residuals to calculate a mean behaviour score for males, virgin females and mated females for each replicate population ($N = 9$). We controlled for effects of the evolutionary sex ratio on variation in aggression among replicate populations by extracting the residuals from linear models of these nine data points against evolutionary sex ratio. We used the residual values to test for correlations in aggression (female headbutts and male lunges, and fencing in both sexes) between males and virgin or mated females. We tested for a correlation between virgin and mated female aggression to assess evidence for a shared genetic basis to female aggression pre- and post-mating.

# 3. Results

## (a) Male aggression and food patch occupancy

We detected no significant influence of the evolutionary sex ratio on the frequency of lunges ($F_{2,6.0} = 1.3$, $p = 0.339$, square root-transformation; figure 1a), the duration of high-intensity aggression (chasing, lunging and tussling; $F_{2,6.0} = 1.4$, $p = 0.322$, log-transformation), or the duration of low-intensity fencing ($F_{2,6.0} = 3.4$, $p = 0.104$, square root-transformation).

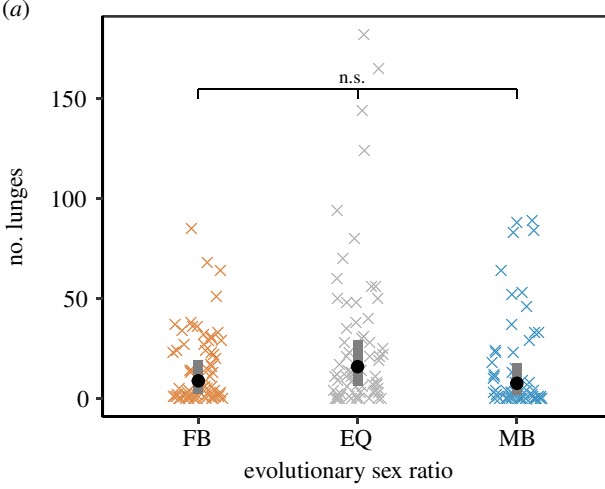

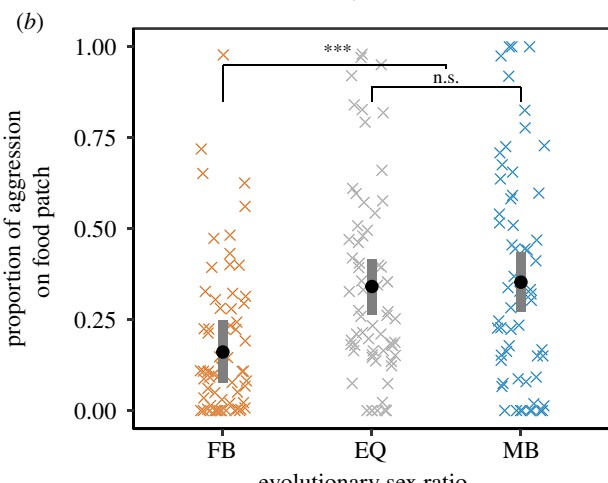

**Figure 1.** Male aggressive behaviour in Experiment 1—'coevolved'. Male aggressive behavior after experimental evolution at female-biased (FB), equal (EQ) or male-biased (MB) sex ratios: lunging (*a*, back-transformed data) and the proportion of aggression performed on food patches (*b*). Circles indicate means. Grey bars indicate 95% confidence intervals; *** indicates $p < 0.001$, * indicates $0.01 < p < 0.05$, n.s. (not significant) indicates $p > 0.05$. (Online version in colour.)

We found that males from female-biased populations spent less time on the food patch compared with male-biased and equal sex ratio populations ($F_{2,5.9} = 14.0$, $p = 0.006$ electronic supplementary material, figure S1B). Males from female-biased populations also performed a lower proportion of total aggression on the food patch relative to males from the other treatments ($\chi_2^2 = 44.7$, $p < 0.001$; figure 1*b*), suggesting differences in resource defence. Aggressive behaviour was related to food patch occupancy. Across all sex ratios, the individual that performed relatively more aggression within a dyad spent relatively more time on the food patch ($\chi_1^2 = 56.5$, $p < 0.001$), and this relationship was weaker as the evolutionary sex ratio became more female-biased ($\chi_2^2 = 113.8$, $p < 0.001$, figure 2*a*). The reduction in food patch use by males from female-biased populations was accompanied by a weak trend towards increased locomotion in these males, relative to those from other sex ratios ($F_{2,6.0} = 4.8$, $p = 0.056$, electronic supplementary material, figure S1A).

## (b) Female aggression and food patch occupancy in Experiment 1—'coevolved'

We found that mating status and evolutionary sex ratio interacted to influence female headbutt frequency (interaction: $F_{2,6.1} = 5.2$, $p = 0.048$; mating status: $F_{1,5.3} = 46.4$, $p < 0.001$; sex ratio: $F_{2,6.1} = 2.0$, $p = 0.213$; figure 3*a*). Headbutting increased after mating in all evolutionary sex ratios, but females from female-biased populations increased headbutting twice as much females from male-biased or equal sex ratio populations (figure 3*a* and electronic supplementary material, table S6). In virgin females, we found no significant effect of evolutionary sex ratio on headbutt frequency ($F_{2,6.1} = 2.7$, $p = 0.149$), but after mating, females from female-biased populations performed more headbutts than females from male-biased populations ($F_{2,6.0} = 5.1$, $p = 0.050$; post-hoc male-biased versus female-biased comparison: $t = 3.2$, d.f. = 6.1, adjusted $p = 0.043$).

There was no evidence of an interaction between mating status and evolutionary sex ratio for female fencing duration, nor evidence for a main effect of evolutionary sex ratio (interaction: $F_{2,6.0} = 2.8$, $p = 0.142$, square root-transformation; sex ratio: $F_{2,5.8} = 3.0$, $p = 0.127$; electronic supplementary material, figure S2A). Fencing duration increased after mating within all evolutionary sex ratios (mating status: $F_{1,6.0} = 42.9$, $p < 0.001$; electronic supplementary material, figure S2A and table S6).

We found no interaction between mating status and evolutionary sex ratio for food patch occupancy, nor a main effect of evolutionary sex ratio (interaction: $F_{2,6.0} = 1.1$, $p = 0.382$; sex ratio: $F_{2,6.0} = 1.4$, $p = 0.312$; electronic supplementary material, figure S2C). Food patch occupancy increased post-mating in all evolutionary sex ratios ($F_{1,5.8} = 15.3$, $p = 0.008$; electronic supplementary material, figure S2C). As in males, the more aggressive mated female within a dyad spent relatively more time occupying the food patch ($\chi_1^2 = 197.5$, $p < 0.001$), with the strongest positive correlation in mated females from male-biased sex ratios (interaction: $\chi_2^2 = 28.4$, $p < 0.001$; sex ratio: $\chi_2^2 = 27.3$, $p < 0.001$; figure 2*b*). However, virgin females showed the opposite pattern: more aggressive virgin females within a dyad spent relatively less time occupying the food patch ($\chi_1^2 = 7.1$, $p = 0.008$), with the strongest negative correlation in male-biased sex ratios (sex ratio: $\chi_2^2 = 15.5$, $p < 0.001$; interaction: $\chi_2^2 = 35.6$, $p < 0.001$; electronic supplementary material, figure S3).

Mating reduced female locomotion ($F_{1,6.0} = 33.6$, $p = 0.001$, square root-transformation; electronic supplementary material, figure S2B), but we detected no influence of evolutionary sex ratio on locomotion, and no interaction between mating and evolutionary sex ratio (evolutionary sex ratio: $F_{2,5.9} = 2.5$, $p = 0.162$; interaction: $F_{2,6.0} = 1.6$, $p = 0.280$).

## (c) Female aggression and food patch occupancy in Experiment 2—'evolved female'

In Experiment 1, the effect of sex ratio on female headbutting might have arisen from evolutionary change in females, from changes in male stimulation of female aggression, or from changes in both sexes. To test whether differences arose from females alone, we mated experimentally evolved females to stock males. As expected, mating caused a general increase in headbutting ($F_{1,6.0} = 10.0$, $p = 0.019$). However, the evolutionary sex ratio did not influence the magnitude of this post-mating increase (evolutionary sex ratio × mating interaction: $F_{2,6.0} = 0.1$, $p = 0.947$, square root-transformation; figure 3*b* and electronic supplementary material, table S6). Females from equal sex ratio populations tended to headbutt more, relative to female-biased and male-biased females ($F_{2,6.0} = 5.0$, $p = 0.053$), regardless of mating status.

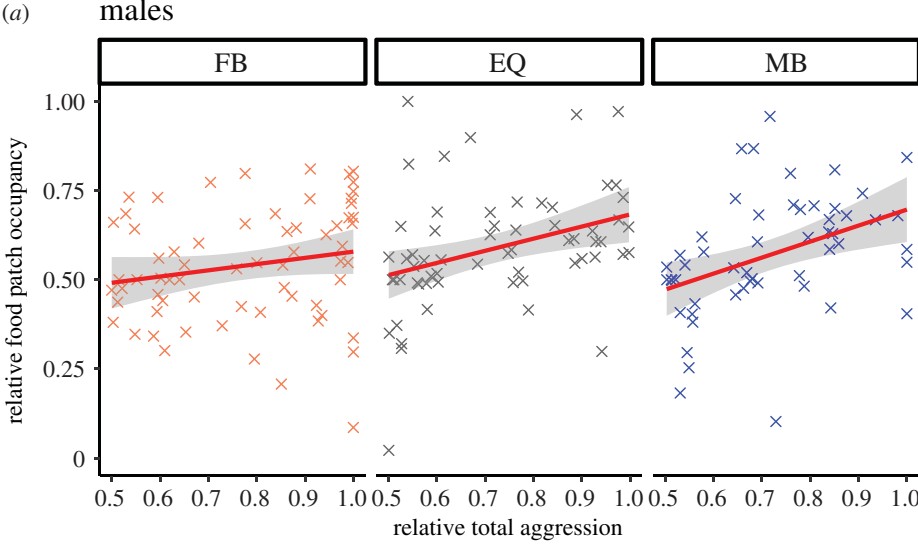

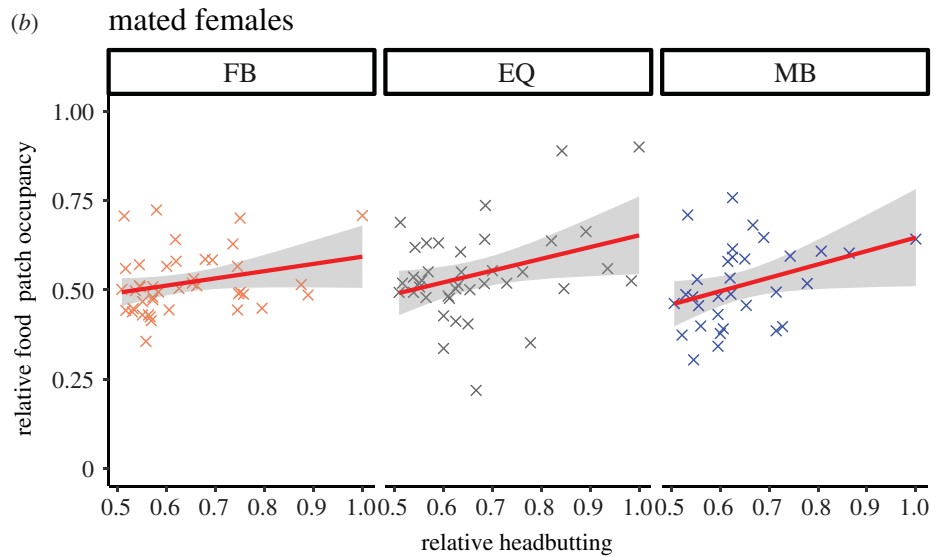

**Figure 2.** The relationship between aggression and food patch occupancy within dyads. The relationship between the proportion of aggression (male total aggression and female headbutts) performed by the most aggressive individual in a pair and the proportion of food patch occupancy for that individual, for males (*a*) and mated females (*b*) at female-biased (FB), equal (EQ) or male-biased (MB) sex ratios. Grey shading indicates 95% confidence intervals. (Online version in colour.)

We observed no significant increase in fencing post-mating ($F_{1,6.1} = 0.1$, $p = 0.745$, log(constant-x)-transformation; electronic supplementary material, figure S4A), in contrast to results from the previous experiment. We found no overall effect of evolutionary sex ratio on female fencing ($F_{2,5.9} = 0.8$, $p = 0.497$), nor an interaction between evolutionary sex ratio and mating ($F_{2,6.0} = 0.6$, $p = 0.559$).

Similar to Experiment 1, we found no interaction between evolutionary sex ratio and mating status for female food patch occupancy ($F_{2,6.0} = 0.6$, $p = 0.601$; electronic supplementary material, figure S4C), nor a main effect of evolutionary sex ratio ($F_{2,5.9} = 1.5$, $p = 0.307$), when evolved females mated with stock males. Mating caused a general increase in food patch occupancy ($F_{1,6.1} = 5.7$, $p = 0.053$).

### (d) Female aggression and food patch occupancy in Experiment 2—'evolved male'

To test whether the differences in female headbutting observed in Experiment 1 were due to evolved differences in male stimulation of female aggression, we mated experimentally evolved males to stock females. All females showed a similar increase in headbutting post-mating

($F_{1,7.9} = 40.2$, $p < 0.001$). There was no effect of male evolutionary sex ratio on headbutt number post-mating ($F_{2,6.1} = 0.4$, $p = 0.706$, figure 3c).

Males did not stimulate a significant increase in fencing in stock females post-mating ($F_{1,7.9} = 0.4$, $p = 0.553$), and we found no effect of male evolutionary sex ratio on female post-mating fencing duration ($F_{2,6.1} = 1.1$, $p = 0.401$; electronic supplementary material, figure S4B).

We detected no interaction between evolutionary sex ratio and mating status on food patch occupancy when stock females mated with experimentally evolved males. Regardless of evolutionary sex ratio, all males stimulated increases in food patch occupancy in stock females post-mating ($F_{1,7.8} = 8.7$, $p = 0.019$), but there was no significant effect of male evolutionary sex ratio on female post-mating food patch occupancy ($F_{2,6.1} = 0.3$, $p = 0.719$; electronic supplementary material, figure S4D).

### (e) The correlation between male and female aggression

We found a positive correlation between the number of male lunges and female headbutts across replicate populations

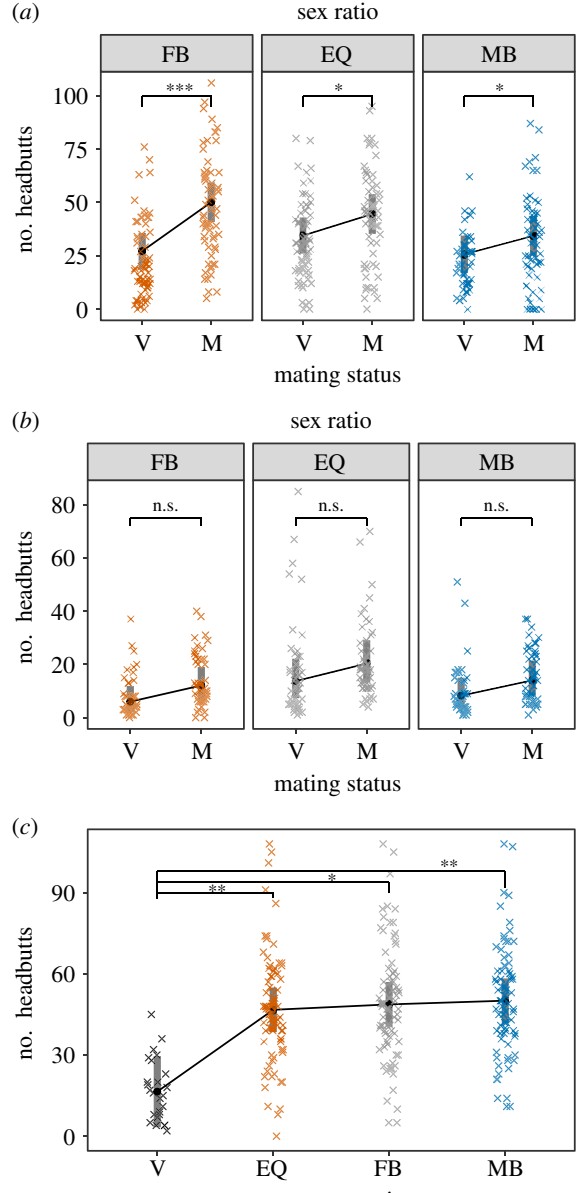

**Figure 3.** Female headbutting. Female headbutting after experimental evolution at female-biased (FB), equal (EQ) or male-biased (MB) sex ratios, for virgin (V) or mated (M) females. Female headbutting was measured when experimentally evolved females mated with experimentally evolved males (*a*; Experiment 1—'coevolved'), when experimentally evolved females mated with stock males (*b*; Experiment 2—'evolved female'; back-transformed data), and when stock females mated with experimentally evolved males (*c*; Experiment 2—'evolved male'). Circles indicate means. Grey bars indicate 95% confidence intervals; *** indicates $p < 0.001$, ** indicates $0.001 < p < 0.01$, * indicates $0.01 < p < 0.05$, n.s. (not significant) indicates $p > 0.05$. (Online version in colour.)

(Spearman's rank correlation, males and virgin females, $\varrho = 0.72$, $S = 34$, $p = 0.037$; males and mated females, $\varrho = 0.63$, $S = 44$, $p = 0.076$; figure 4*a,b*), but found no correlation in fencing duration between the sexes (males and virgin females, $\varrho = -0.02$, $S = 122$, $p = 0.982$; males and mated females, $\varrho = -0.25$, $S = 150$, $p = 0.521$).

## (f) The correlation between virgin and mated female aggression

We found a positive correlation between pre- and post-mating female headbutting frequency across replicate populations

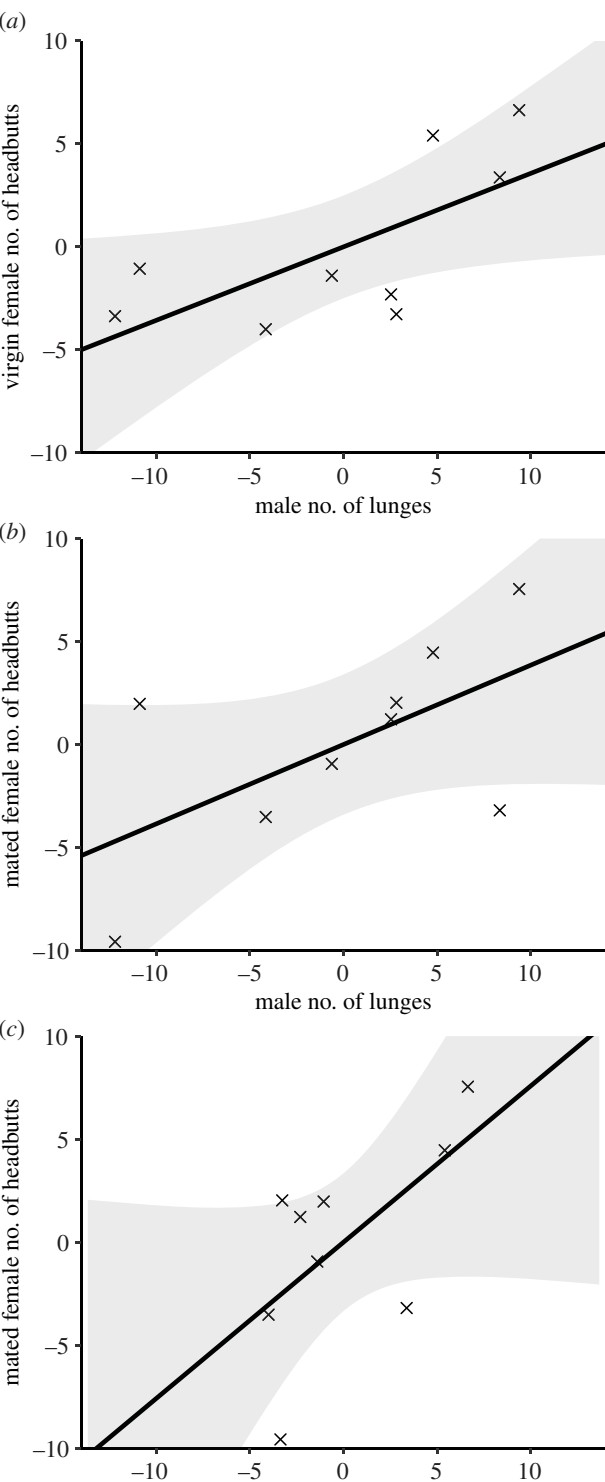

**Figure 4.** Correlations between male and female aggressive behaviours. The relationship between male and female aggressive behaviour (male lunges and headbutts by virgin (*a*) or mated females (*b*)) and between virgin and mated female headbutts (*c*). Points are residual values from models controlling for day, time and sex ratio. Lines indicate the monotonic fit from Spearman's correlation; grey shading indicates the 95% confidence interval.

(Spearman's rank correlation, $\varrho = 0.70$, $S = 36$, $p = 0.043$, figure 4*c*), but found no correlation in fencing behaviour ($\varrho = 0.07$, $S = 112$, $p = 0.880$).

## 4. Discussion

We investigated how aggression evolves in response to the intensity of intra-sexual competition by assaying aggression after

experimentally manipulating the population sex ratio for greater than 75 generations. We predicted that males and females would evolve increased aggression after evolution in populations biased towards their sex, and our results support this prediction strongly in females and weakly in males. We observed a greater increase in aggression after mating in females from female-biased populations, as predicted if higher post-mating aggression is adaptive for females. Surprisingly, differences in the magnitude of this increase among sex ratios occurred only after matings between experimentally evolved males and females, and not when experimentally evolved individuals mated with stock flies. These results suggest that differences in the post-mating increase in aggression do not arise through evolution in either sex independently, but might depend on coevolved interactions between the sexes. We found positive correlations in aggression between the sexes, consistent with a shared genetic basis for aggression. Our results suggest that the intensity of competition can determine the strength of sexual and social selection on aspects of aggression and food patch occupancy in both male and female *D. melanogaster*, shaping the evolution of these behaviours.

## (a) The evolution of male aggression with sex ratio

We predicted that evolution under stronger sexual selection, through more intense competition for mates in male-biased populations, should lead to increased male aggression, mirroring plastic changes in response to sex ratio in a wide range of species [14,15,49]. The results offer only weak support for this prediction. On the one hand, the absence of evolved differences in the frequency and duration of male aggression in response to sex ratio does not support the prediction. Two possible explanations for the absence of response are that selection favours plasticity in aggression rather than fixed increases or decreases [50]; or that changes in the strength of competition for mates with sex ratio are balanced by changes in rival density and costs of fighting [10,51–53]. However, neither hypothesis accounts for our observations of sex ratio effects on the evolution of female aggression and male aggression in relation to food patches.

On the other hand, we observed the evolution of reduced food patch occupancy, a reduced proportion of aggression performed on food, and a weaker relationship between aggression and food occupancy, in males from female-biased populations relative to other males. The function of male aggression in gaining access to food resources is supported both by our finding that more aggressive males spend relatively more time occupying the food patch, and by previous reports that aggressive male *D. melanogaster* win access to food patches [54,55], which increases their access to mates [16,40,55]. Our results are consistent with weaker selection for the use of aggression to attain access to food patches under female-biased conditions, in which weaker competition for mates is expected to reduce the benefits of dominating breeding sites [15,56]. An alternative hypothesis is that reduced male food patch occupancy after evolution in female-biased populations might reflect reduced female aggregation on food patches. However, females aggregate more, not less, on food patches in our female-biased populations [37].

## (b) The evolution of female aggression with sex ratio

Females increase aggression after mating in many species [20,24,25,33,35]. Our results are consistent with this pattern. Increased aggression post-mating might represent an adaptive response that relates to the acquisition or defence of nutritional resources required for reproduction, as the switch to a post-mating reproductive state increases female feeding and protein requirements [29,57,58]. Our findings that females from all sex ratio treatments display increased food patch occupation post-mating, and that aggression is positively related to food occupancy in mated females, support this idea.

We found that the evolutionary sex ratio influences both the level of aggression in mated females and the magnitude of the post-mating increase in aggression, with more headbutts and a greater increase in headbutt frequency post-mating in females from female-biased populations. The greater intensity of female competition in female-biased populations might impose stronger selection favouring aggression in the nutritionally demanding mated state. Our results suggest that the intensity of intra-sexual competition can shape the evolution of female aggression, and that this might relate to nutritional defence, although causality in this relationship is unclear. Future work testing the relationship between female aggression, defence of food and reproductive success would improve understanding of the function of aggression in this species.

Our findings are inconsistent with the hypotheses that evolution in either sex alone explains the observed effect of sex ratio on the female post-mating increase in aggression. Previous work has demonstrated that the receipt of male sperm and the seminal fluid protein 'sex peptide' directly influence female aggression in *D. melanogaster* [35]. Moreover, some properties of the male ejaculate such as sperm competitiveness and ejaculate expenditure show evolvability in response to the sex ratio [27,59–61]. However, a male's ability to stimulate female aggression did not appear to evolve in the conditions of our experiment.

We are left with the hypothesis that the female post-mating behaviours observed when both sexes had experimentally evolved reflect coevolved interactions between the sexes, such that evolved changes occur only after matings between individuals from the same social environment. Similar complex interactions between male and female genotypes are known in *Drosophila*. For example, the effect of some male sex peptide alleles on sperm competitiveness depends on the female sex peptide receptor allele [62]. Likewise, sperm success can depend on interactions between male and female genotypes [63]. Although we know that female post-mating aggression is linked to the receipt of male ejaculates [35], the downstream mechanism within females remains elusive. Research into the post-mating regulation of female aggression would help further evaluation of the coevolution hypothesis.

## (c) A positive correlation in aggression between the sexes

Studying the evolution of male and female aggression simultaneously allowed us to evaluate the hypothesis that aggression is genetically correlated between the sexes. This is especially relevant because female aggression has sometimes been considered a non-adaptive by-product of selection for male aggression [4,64] and has only recently been studied as an adaptive female trait [21].

Our observation of a positive correlation between male lunging and female headbutting across replicate populations is consistent with a shared genetic basis for aggression. There is evidence that selection for aggression in male *D. melanogaster* results in correlated responses in female

aggression [65], supporting this idea. This suggests the possibility that genetic constraints might impede the evolution of sex-specific optimal aggression. However, our observation of divergent responses to sex ratio for males and females suggests that a genetic correlation for aggression does not completely restrict its independent evolution in each sex. Alternatively, a positive correlation could arise if aggression forms a behavioural syndrome with other coevolving intersexual behaviours, such as male harassment of females and female resistance. However, this seems unlikely because there is little evidence that aggression covaries across contexts in *D. melanogaster* [66] and intra-sexual aggressive behaviours are rarely directed at the opposite sex [67]. Furthermore, the positive correlation between headbutting by virgin and mated females suggests a consistent genetic basis for female aggression pre- and post-mating, such that females have a baseline level of aggression that is enhanced by mating. By contrast, the absence of correlations in fencing behaviour between males and females, and between virgin and mated females, across replicate populations might reflect differences in the function of this low-intensity aggressive behaviour between the sexes, and within females depending on their mating status. Fencing is performed by both sexes, but there are distinct differences in the aggressive strategies of males and females [33] and in females pre- and post-mating [35]. If there are distinct genetic pathways underlying low- and high-intensity aggression, then the extent to which sex-specific aggression is constrained by a shared genetic basis may vary for different aggressive behaviours.

Our study provides evidence that the strength of sexual and social selection, mediated by competition for mates and resources, can shape the evolution of aggressive behaviours in both male and female *D. melanogaster*. These effects differ between the sexes, which might reflect different routes by which aggression influences reproductive success [2]. The higher energy demands of reproduction in females might result in greater reproductive costs from energetically expensive aggression in females than in males, causing reduced female aggression with greater sensitivity to the ecological setting.

Furthermore, although we found evidence consistent with a shared genetic basis for aggression, our observation of divergent responses to sex ratio for males and females suggests that a genetic correlation for aggression does not completely restrict its independent evolution. Our study also highlights that increased female aggression in response to mating might be sensitive to adaptations in both sexes. This underscores the value of future study of the mechanisms underlying the female post-mating increase in aggression, and of studying behaviour in both sexes.

Data accessibility. Data are available from the Oxford University Research Archive (ORA) at: https://ora.ox.ac.uk/objects/uuid:27e23 e33-8101-4469-8f9e-982c5ae8754d.

Authors' contributions. E.B., D.E., S.W. and J.C.P. conceived the ideas and designed the methodology; J.N., C.A. and L.H. assisted in designing methodology; T.C. and S.W. designed the experimental evolution protocol; T.C. and W.G.R. performed the experimental evolution; E.B., D.E., J.N., C.A. and L.H. collected the data; E.B., D.E. and J.C.P. analysed the data; E.B., D.E. and J.C.P. drafted the initial version of the manuscript and all authors contributed to later versions of the manuscript.

Competing interests. We declare we have no competing interests.

Funding. E.B. was funded by a fellowship from Christ Church College and grants from the John Fell Fund (University of Oxford, ATD12830) and the Association for the Study of Animal Behaviour. D.E. was funded by the Biotechnology and Biological Sciences Research Council (BBSRC) Doctoral Training Partnership. T.C. was funded by the Natural Environment Research Council (NERC; NE/K004697/1). W.G.R. was funded by a NERC grant to T.C. (NE/R010056/1). S.W. was funded by a BBSRC David Phillips Fellowship (BB/K014544/1). J.C.P. was funded by an NERC fellowship (NE/P017193/1).

Acknowledgements. We thank two anonymous reviewers for comments that improved the manuscript.

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
