## [Peer Review File · Proceedings of the Royal Society B: Biological Sciences]

Review History

RSPB-2020-1770.R0 (Original submission)

Review form: Reviewer 1

Recommendation

Accept with minor revision (please list in comments)

Scientific importance: Is the manuscript an original and important contribution to its field?
Excellent

General interest: Is the paper of sufficient general interest?
Excellent

Quality of the paper: Is the overall quality of the paper suitable?
Good

Is the length of the paper justified?
Yes

Should the paper be seen by a specialist statistical reviewer?
No

Do you have any concerns about statistical analyses in this paper? If so, please specify them explicitly in your report.

No

It is a condition of publication that authors make their supporting data, code and materials available - either as supplementary material or hosted in an external repository. Please rate, if applicable, the supporting data on the following criteria.

Is it accessible?

Yes

Is it clear?

Yes

Is it adequate?

Yes

Do you have any ethical concerns with this paper?

No

Comments to the Author

This is a really interesting paper looking at how evolving under different sex ratios (and thus different levels of intrasexual competition) influences the evolution of male and female aggression behaviour in *Drosophila melanogaster*. The authors use well established experimental evolution lines and conduct a series of experiments/assays to get at this question from several angles and in so doing provide interesting and novel data on the topic of what drives the evolution of aggressive behaviour. For the most part the manuscript is clear and well written (although I do think it could be polished further for readability), the data collection and analysis methods are sound (although more detail and restructuring is needed in some areas) and the synthesis of literature and interpretation of results is balanced.

Most of the following detailed comments are relatively minor and should be easily addressed by the authors. They are either aimed at improving the clarity of the manuscript, driven by my interest in the study, or are simply typos that need correcting (sorry I can't help myself). The comments appear in the order they found in the manuscript

Line 26: add "s" to impact

Line 27: I think it is worth early on stating what you mean by competition or being more explicit by what type of competition you are referring to. Here I first thought you simply meant competition over food resources. In the abstract here it might be enough to say "intrasexual" competition, or the term you use in the intro "reproductive competition".

Line 34: replace "stronger" with "greater"

Line 35: Which effect do you mean by "This effect". Both the male and female responses to selection or just that of the female? Be explicit.

Line 35-37: I find this sentence a little awkward. Also I'm not sure from your experiment whether you can invoke male stimulation of females via ejaculates as what drives coevolution. There could be other aspects of mating that drive this pattern. I guess previous evidence from *drosophila* might suggest this is the most likely pathway, but without that context in your abstract I would stick with what this study specifically can tell you, which is simply that the response is likely the result of coevolution.

Line 37: where you say "our results" I think you are now talking about a different result. To make this clearer I would state the result explicitly or make it clearer you are now talking about something new by stating something like "we also found evidence for..."

Line 70-80: This paragraph presents an interesting idea, but I found it really difficult at first to figure out what you are getting at. Are you basically saying that males could manipulate females to show increased aggression after mating because they benefit from it (ie more eggs fertilised by

them or eggs that are fertilised by them have more resources due to increased female access to food) without experiencing potential costs (ie injury)? That's cool, but I guess they only don't suffer the costs if the costs are delayed... and I wonder if it's more likely a side-effect eg males transfer something that increases female egg laying rate and this means they need greater access to food, thus have increased motivation to fight for it... Ok well the later part of this comment is just musings, but I do think the point of this paragraph could be made clearer.

Line 85: preface competition with "intrasexual" or "reproductive" to avoid confusion

Line 87: you say it later in the manuscript, but I think it is worth mentioning here, that males don't eat much as adults.

Line 91: The last sentence of this paragraph is awkward perhaps change to something like "Therefore differences in female aggression could represent a response to evolved differences in male ejaculates as well as the direct evolution of female behaviour."

Line 96-99: This sentence needs revising for flow and clarity

Line 101: weird to have a sentence starting with "and" after a sentence starting with "finally".

Also this last sentence sounds tacked on and needs to be integrated better

Line 108-114: I found this description of the experiments really confusing. Rather than try and explain why I thought it was confusing. Here is a suggestion for how I think it could be made less confusing. Obviously I don't expect you to use my words, and I might have some of the details not quite right... but I think something like this would make the structure and purpose of your experiments clearer.

"We conducted two experiments in the first we measured intra-sexual aggression in virgin females, mated females and mated male that had evolved under male biased, equal and female biased evolutionary sex ratios. In this experiment all flies were mated with partners from the same replicate line. We tested both mated and virgin females because it is known that females show increased aggression after mating. However we only tested mated males, because they show no such change in behaviour. We then conducted a second experiment designed to look whether differences in female behaviour reflected direct evolutionary responses of aggression or reflected changes that resulted from the evolution of male traits. This second experiment involved mating females from our ESR lines to stock males as well as stock females to male from our ESR lines and measuring female aggression both before and after mating."

Line 133: What did you do with them after the trial? Put them back in individual containers?

Line 134: so virgin females were different individuals to mated females? ie they weren't the same females measured before and after mating? I guess that means virgins and mated females were tested at the same age? Could provide more detail and be more explicit around these points.

Line 140: for females - were mated females always paired with mated females and virgin with virgin? I wonder if pairing with replicates could exacerbate differences between selection regimes due to escalation of contests??? Would it have been better to pair everyone with a standard competitor?

Line 147: delete "in"

Line 150-165: Generally I think this paragraph could be much more succinct. More detail in following comments

Line 157: I'm not sure how you sum the amount of time spent lunging since this is a count?

Line 157: description of fencing behaviours could be incorporated into previous sentence using brackets

Line 160: The average duration of time spent on a food patch seems like a strange proxy for male territoriality to me. If both males spent a lot of time on the food patch.

Line 161: put info that males spend little time feeding in the intro.

Line 163-165: Why only in the coevolved experiment? what did you do in the other experiment? why only measure a subset of trials for low intensity? You have videos for all, and sample sizes aren't that much different, seems like a strange decision that should be justified.

Line 172-210: While your data analysis seems largely sound, the description is confusing and incomplete, this section needs a thorough revision. In so doing I also think it could be made more concise. Perhaps structuring it in a similar way to your results section would help? Ie coevolved exp - males, females, female evolved experiment, male evolved experiment

172-177: is this just the "coevolved" experiment? what traits did you analyse?

Line 176: Why include observer for females and not males?

Line 177-179: Why not analyse these in the same model so you can look for an interaction? Also what traits were you looking at here?

Line 185-187: is this in addition to the lmm described above?

Line 187: where do you describe what you did for the “evolved female” experiment

Line 188-191: Did you describe analysis of males above? I would put all male analyses together

Line 194: add “s” to population

Line 195: what determines whether including observer is applicable or not?

Line 207-210: I wonder what this analysis adds to your paper – you don’t really talk about it later as far as I can tell. Perhaps you could put it in the supplement and just say above where you look at aggression that you did not include mating behaviour as a covariate, because it did not differ? Although, I must say I’m a little surprised it doesn’t differ, is this consistent with previous studies on these lines? If you think these analyses are important then you need to add more detail of model setup.

Line 212: throughout results be consistent in the number of decimal places (standard is 3) for reporting p values.

Line 223-224: This suggests to me that they are not really defending this resource as you suggest earlier. I wonder whether it is worth somehow taking into account the amount of space in the vial that it takes up to show that they actually defend it more than you would expect by chance? Or whether you need to think about your interpretation of the resource?

Line 250: do you mean it was “lower” in EQ females? or that it reduced after mating in these females?

Line 315: manipulated populations sex ratio... and then what?

Line 317: “after evolving in a sex ratio skewed toward their own sex”

Line 317: here and elsewhere I find the phrase “evolve changes” weird. Consider rewording

Line 322: “females” rather than individuals?

Line 343: I guess I’m left wondering, if this is true, why don’t they spend more time defending it?

Line 344: Could it be that decreased male aggression and territoriality is not directly a result of weaker male male competition, but that more females in this selection regime means that there is less female aggregation around the resource... ie some females are booted off, and so the food bottle becomes less valuable to defend. Would be nice to see some data on behaviour from the actual experimental lines if you have it... although not necessary for publication.

Line 346-361: This is an interesting argument but it a bit vague and I’m left wondering exactly what the predictions are you would expect for low and high intensity aggression for each of the ERS’s, and whether your results fit those predictions.

Line 366: delete “s” from findings

Line 368: You don’t really discuss reasons differences between male and female responses in detail, but I wonder whether they could arise due to difference in the resource they are competing for?

Line 373-376: I’m not entirely sure that food patch occupancy actually reflects nutritional defence. If I remember correctly you averaged the time two females from the same line spent on the food patch so high values mean that both females spent lots of time on the patch, which would indicate they aren’t defending it? But instead the opposite.

Line 389-390: What is the reasoning/evidence behind this statement?

Line 394: delete “s” from males

Line 391-400: In your experiment you only mated pairs within their replicate. I wonder if you had performed crosses across lines but within selection regimes? You could have gotten a better handle on whether this result was due to coevolution per se or something to do with male and/or female traits?

Figure 3: Would be worth mentioning in figure caption that data points reflect mean values for each replicate (or whatever it is they represent)

Review form: Reviewer 2

Recommendation

Accept with minor revision (please list in comments)

Scientific importance: Is the manuscript an original and important contribution to its field?

Good

General interest: Is the paper of sufficient general interest?

Excellent

Quality of the paper: Is the overall quality of the paper suitable?

Excellent

Is the length of the paper justified?

Yes

Should the paper be seen by a specialist statistical reviewer?

No

Do you have any concerns about statistical analyses in this paper? If so, please specify them explicitly in your report.

No

It is a condition of publication that authors make their supporting data, code and materials available - either as supplementary material or hosted in an external repository. Please rate, if applicable, the supporting data on the following criteria.

Is it accessible?

Yes

Is it clear?

Yes

Is it adequate?

Yes

Do you have any ethical concerns with this paper?

No

Comments to the Author

This paper explored the evolution of aggression in fruit flies by manipulating the sex ratio of populations for >75 generations. The authors tested the hypothesis that the overabundant sex should experience greater intra-sexual competition for resources and/or mates, and therefore evolve greater aggression. Interestingly, these hypotheses were supported only when flies mated with other members of their own population. This suggests that some kind of co-evolutionary process may have occurred, which is fascinating and perhaps unexpected.

The manuscript is in pretty good shape as it stands, and I think will make a nice contribution to Proc B. I have a few concerns that I think the authors will be able to resolve fairly easily. From most important to least important, they are:

1. I didn't understand the statistical approach used to distinguish between underlying inter-sexual genetic correlations and co-evolutionary processes, and this seems really important. Something about residuals...? Is this the equivalent of examining the inter-sexual correlation

within populations, and if so, why not do that directly? This needs to be explained and justified more clearly.

Also, a description of these 2 hypotheses (shared underlying genetics and co-evolution) isn't given until the methods section (around line 200), but it should be in the introduction (unless I missed it, if so sorry).

2. No evidence is provided that the effect of the stock population individuals would be generalizable across all "non-coevolved" individuals, but this is central to the interpretation of the results. i.e., using a different stock population could have produced different results. I think the authors need to add this as a caveat to the discussion section

3. More needs to be explained about the ecological conditions in which selection occurred. This wasn't mentioned until the discussion, but I was wondering about it when reading the results for patch occupancy.

4. Acronyms make the paper much harder to read and are not necessary when they just stand for 2 words. Each time I saw "FB" and the others I had to pause and try to remember what it means. Please switch these to real words.

Decision letter (RSPB-2020-1770.R0)

04-Sep-2020

Dear Dr Bath:

I am writing to inform you that we have received referees' reports on your manuscript RSPB-2020-1770 entitled "Sex ratio and the evolution of aggression in fruit flies".

Whilst the referees, the Associate Editor and I are all enthusiastic about the manuscript, several aspects have been identified which, taken together, mean that substantial revision of the paper is required. The manuscript has therefore been rejected in its current form. However we would be happy to consider a resubmission, provided the comments of the referees are fully addressed; please see the very thorough referees' reports below. Please note also that this is not a provisional acceptance.

Sincerely,
 Professor Loeske Kruuk
 mailto: proceedingsb@royalsociety.org

Associate Editor

Board Member: 1

Comments to Author:

This study is very interesting. The authors have evolved populations of *D. melanogaster* under different operational sex ratios (thus manipulating the intensity of sexual selection), and looked at corresponding changes in the frequency of male and female aggression.

The paper was sent to two expert referees, each of whom enjoyed the study. Referee 1 found some of the description of the experimental design and behavioural measures ambiguous, and requiring of clarification. They have provided some helpful suggestions, as well as a series of detailed comments requiring attention. Referee 2 found it difficult to understand the statistical analyses described from line 199. I also struggle to follow what you have done, and why, and this needs to be explained more clearly. Referee 2 has also made other insightful comments, each requiring attention.

I also note that your linear mixed models have utilised Wald Chi-square tests, which are regarded as anti-conservative, and not ideal. Much better to apply Kenward Rogers or Satterthwaite F tests, which are less likely to give you Type 1 error; another massive benefit is you can then see the denominator degrees of freedom to make sure that you have specified the random statement correctly [i.e. if your denominator dfs blow out beyond your number of replicates, you know you have a problem and inherent pseudoreplication (see Arnqvist, G. (2020). "Mixed Models Offer No Freedom from Degrees of Freedom." *Trends in Ecology & Evolution*.)]. Alternative is to use a parametric bootstrap. Surprisingly many models are inherently pseudoreplicated in ecology and evolution because the random intercept fails to account for the correct level of analysis for fixed interactions - this often requires fitting with random slopes.

Reviewer(s)' Comments to Author:

Referee: 1

Comments to the Author(s)

This is a really interesting paper looking at how evolving under different sex ratios (and thus different levels of intrasexual competition) influences the evolution of male and female aggression behaviour in *Drosophila melanogaster*. The authors use well established experimental evolution lines and conduct a series of experiments/assays to get at this question from several angles and in so doing provide interesting and novel data on the topic of what drives the evolution of aggressive behaviour. For the most part the manuscript is clear and well written (although I do think it could be polished further for readability), the data collection and analysis methods are sound (although more detail and restructuring is needed in some areas) and the synthesis of literature and interpretation of results is balanced.

Most of the following detailed comments are relatively minor and should be easily addressed by the authors. They are either aimed at improving the clarity of the manuscript, driven by my interest in the study, or are simply typos that need correcting (sorry I can't help myself). The comments appear in the order they found in the manuscript

Line 26: add "s" to impact

Line 27: I think it is worth early on stating what you mean by competition or being more explicit by what type of competition you are referring to. Here I first thought you simply meant competition over food resources. In the abstract here it might be enough to say "intrasexual" competition, or the term you use in the intro "reproductive competition".

Line 34: replace "stronger" with "greater"

Line 35: Which effect do you mean by "This effect". Both the male and female responses to selection or just that of the female? Be explicit.

Line 35-37: I find this sentence a little awkward. Also I'm not sure from your experiment whether you can invoke male stimulation of females via ejaculates as what drives coevolution. There could be other aspects of mating that drive this pattern. I guess previous evidence from drosophila might suggest this is the most likely pathway, but without that context in your abstract I would stick with what this study specifically can tell you, which is simply that the response is likely the result of coevolution.

Line 37: where you say "our results" I think you are now talking about a different result. To make this clearer I would state the result explicitly or make it clearer you are now talking about something new by stating something like "we also found evidence for..."

Line 70-80: This paragraph presents an interesting idea, but I found it really difficult at first to figure out what you are getting at. Are you basically saying that males could manipulate females to show increased aggression after mating because they benefit from it (ie more eggs fertilised by them or eggs that are fertilised by them have more resources due to increased female access to food) without experiencing potential costs (ie injury)? That's cool, but I guess they only don't suffer the costs if the costs are delayed... and I wonder if it's more likely a side-effect eg males transfer something that increases female egg laying rate and this means they need greater access to food, thus have increased motivation to fight for it... Ok well the later part of this comment is just musings, but I do think the point of this paragraph could be made clearer.

Line 85: preface competition with "intrasexual" or "reproductive" to avoid confusion

Line 87: you say it later in the manuscript, but I think it is worth mentioning here, that males don't eat much as adults.

Line 91: The last sentence of this paragraph is awkward perhaps change to something like "Therefore differences in female aggression could represent a response to evolved differences in male ejaculates as well as the direct evolution of female behaviour."

Line 96-99: This sentence needs revising for flow and clarity

Line 101: weird to have a sentence starting with "and" after a sentence starting with "finally".

Also this last sentence sounds tacked on and needs to be integrated better

Line 108-114: I found this description of the experiments really confusing. Rather than try and explain why I thought it was confusing. Here is a suggestion for how I think it could be made less confusing. Obviously I don't expect you to use my words, and I might have some of the details not quite right... but I think something like this would make the structure and purpose of your experiments clearer.

"We conducted two experiments in the first we measured intra-sexual aggression in virgin females, mated females and mated male that had evolved under male biased, equal and female biased evolutionary sex ratios. In this experiment all flies were mated with partners from the same replicate line. We tested both mated and virgin females because it is known that females show increased aggression after mating. However we only tested mated males, because they show no such change in behaviour. We then conducted a second experiment designed to look whether differences in female behaviour reflected direct evolutionary responses of aggression or reflected changes that resulted from the evolution of male traits. This second experiment involved mating females from our ESR lines to stock males as well as stock females to male from our ESR lines and measuring female aggression both before and after mating."

Line 133: What did you do with them after the trial? Put them back in individual containers?

Line 134: so virgin females were different individuals to mated females? ie they weren't the same females measured before and after mating? I guess that means virgins and mated females were tested at the same age? Could provide more detail and be more explicit around these points.

Line 140: for females - were mated females always paired with mated females and virgin with virgin? I wonder if pairing with replicates could exacerbate differences between selection regimes

due to escalation of contests??? Would it have been better to pair everyone with a standard competitor?

Line 147: delete "in"

Line 150-165: Generally I think this paragraph could be much more succinct. More detail in following comments

Line 157: I'm not sure how you sum the amount of time spent lunging since this is a count?

Line 157: description of fencing behaviours could be incorporated into previous sentence using brackets

Line 160: The average duration of time spent on a food patch seems like a strange proxy for male territoriality to me. If both males spent a lot of time on the food patch .

Line 161: put info that males spend little time feeding in the intro.

Line 163-165: Why only in the coevolved experiment? what did you do in the other experiment? why only measure a subset of trials for low intensity? You have videos for all, and sample sizes aren't that much different, seems like a strange decision that should be justified.

Line 172-210: While your data analysis seems largely sound, the description is confusing and incomplete, this section needs a thorough revision. In so doing I also think it could be made more concise. Perhaps structuring it in a similar way to your results section would help? I.e coevolved exp - males, females, female evolved experiment, male evolved experiment

172-177: is this just the "coevolved" experiment? what traits did you analyse?

Line 176: Why include observer for females and not males?

Line 177-179: Why not analyse these in the same model so you can look for an interaction? Also what traits were you looking at here?

Line 185-187: is this in addition to the lmm described above?

Line 187: where do you describe what you did for the "evolved female" experiment

Line 188-191: Did you describe analysis of males above? I would put all male analyses together

Line 194: add "s" to population

Line 195: what determines whether including observer is applicable or not?

Line 207-210: I wonder what this analysis adds to your paper - you don't really talk about it later as far as I can tell. Perhaps you could put it in the supplement and just say above where you look at aggression that you did not include mating behaviour as a covariate, because it did not differ? Although, I must say I'm a little surprised it doesn't differ, is this consistent with previous studies on these lines? If you think these analyses are important then you need to add more detail of model setup.

Line 212: throughout results be consistent in the number of decimal places (standard is 3) for reporting p values.

Line 223-224: This suggests to me that they are not really defending this resource as you suggest earlier. I wonder whether it is worth somehow taking into account the amount of space in the vial that it takes up to show that they actually defend it more than you would expect by chance? Or whether you need to think about your interpretation of the resource?

Line 250: do you mean it was "lower" in EQ females? or that it reduced after mating in these females?

Line 315: manipulated populations sex ratio... and then what?

Line 317: "after evolving in a sex ratio skewed toward their own sex"

Line 317: here and elsewhere I find the phrase "evolve changes" weird. Consider rewording

Line 322: "females" rather than individuals?

Line 343: I guess I'm left wondering, if this is true, why don't they spend more time defending it?

Line 344: Could it be that decreased male aggression and territoriality is not directly a result of weaker male competition, but that more females in this selection regime means that there is less female aggregation around the resource... ie some females are booted off, and so the food bottle becomes less valuable to defend. Would be nice to see some data on behaviour from the actual experimental lines if you have it... although not necessary for publication.

Line 346-361: This is an interesting argument but it a bit vague and I'm left wondering exactly what the predictions are you would expect for low and high intensity aggression for each of the ERS's, and whether your results fit those predictions.

Line 366: delete "s" from findings

Line 368: You don't really discuss reasons differences between male and female responses in detail, but I wonder whether they could arise due to difference in the resource they are competing for?

Line 373-376: I'm not entirely sure that food patch occupancy actually reflects nutritional defence. If I remember correctly you averaged the time two females from the same line spent on the food patch so high values mean that both females spent lots of time on the patch, which would indicate they aren't defending it? But instead the opposite.

Line 389-390: What is the reasoning/evidence behind this statement?

Line 394: delete "s" from males

Line 391-400: In your experiment you only mated pairs within their replicate. I wonder if you had performed crosses across lines but within selection regimes? You could have gotten a better handle on whether this result was due to coevolution per se or something to do with male and/or female traits?

Figure 3: Would be worth mentioning in figure caption that data points reflect mean values for each replicate (or whatever it is they represent)

Referee: 2

Comments to the Author(s)

This paper explored the evolution of aggression in fruit flies by manipulating the sex ratio of populations for >75 generations. The authors tested the hypothesis that the overabundant sex should experience greater intra-sexual competition for resources and/or mates, and therefore evolve greater aggression. Interestingly, these hypotheses were supported only when flies mated with other members of their own population. This suggests that some kind of co-evolutionary process may have occurred, which is fascinating and perhaps unexpected.

The manuscript is in pretty good shape as it stands, and I think will make a nice contribution to Proc B. I have a few concerns that I think the authors will be able to resolve fairly easily. From most important to least important, they are:

1. I didn't understand the statistical approach used to distinguish between underlying inter-sexual genetic correlations and co-evolutionary processes, and this seems really important. Something about residuals...? Is this the equivalent of examining the inter-sexual correlation within populations, and if so, why not do that directly? This needs to be explained and justified more clearly.

Also, a description of these 2 hypotheses (shared underlying genetics and co-evolution) isn't given until the methods section (around line 200), but it should be in the introduction (unless I missed it, if so sorry).

2. No evidence is provided that the effect of the stock population individuals would be generalizable across all "non-coevolved" individuals, but this is central to the interpretation of the results. i.e., using a different stock population could have produced different results. I think the authors need to add this as a caveat to the discussion section

3. More needs to be explained about the ecological conditions in which selection occurred. This wasn't mentioned until the discussion, but I was wondering about it when reading the results for patch occupancy.

4. Acronyms make the paper much harder to read and are not necessary when they just stand for 2 words. Each time I saw "FB" and the others I had to pause and try to remember what it means. Please switch these to real words.

Author's Response to Decision Letter for (RSPB-2020-1770.R0)

See Appendix A.

RSPB-2020-3053.R0

Review form: Reviewer 2

Recommendation

Accept as is

Scientific importance: Is the manuscript an original and important contribution to its field?

Good

General interest: Is the paper of sufficient general interest?

Good

Quality of the paper: Is the overall quality of the paper suitable?

Excellent

Is the length of the paper justified?

Yes

Should the paper be seen by a specialist statistical reviewer?

No

Do you have any concerns about statistical analyses in this paper? If so, please specify them explicitly in your report.

No

It is a condition of publication that authors make their supporting data, code and materials available - either as supplementary material or hosted in an external repository. Please rate, if applicable, the supporting data on the following criteria.

Is it accessible?

Yes

Is it clear?

Yes

Is it adequate?

Yes

Do you have any ethical concerns with this paper?

No

Comments to the Author

The authors have done a good job of addressing my comments from the initial round of reviews, and the manuscript is now much clearer. I have a few small additional comments, listed below, that are not sufficient to warrant a full revision but could further improve the final manuscript. I look forward to seeing this paper in Proc B.

Lines 41-42 seem overly general, e.g., males may compete for territories, females may compete for reproductive opportunities, etc.

Lines 53-55 is there a specific idea here about what might co-vary with competition that is expected to produce increased aggression?

Line 116 just to be precise, there is evidence for aggression as a strategy for mate guarding in flies (<https://www.sciencedirect.com/science/article/abs/pii/S0003347215003206>). I don't think this change in aggression following mating would be relevant to your experimental design, however, since the males were separated from their mates during the aggression trials.

Decision letter (RSPB-2020-3053.R0)

22-Jan-2021

Dear Dr Bath,

Thank you for the revised version of this manuscript - it's a really fascinating set of results. The new version has now been reviewed by one of the original reviewers, who was very happy with the revisions but has suggested a few changes. Thank you in particular for incorporating the F test and d.f. information, which has been very useful for clarifying the model structure. However, as you'll see from the Associate Editor's comments below, it has indicated a potential issue of concern - though this is something that is easily fixed. In my discussions with the AE over the paper, we were both aware of finding it quite difficult to navigate the output of the models, so we would also like you to include some more information on the model output in tables (see detailed comments below). The reviewer's, the AE's and my comments are all included at the end of this email for your reference, and we invite you to revise your manuscript to address them.

Research ethics:

Use of animals and field studies:

It is a condition of publication that you make available the data and research materials supporting the results in the article (<https://royalsociety.org/journals/authors/author-guidelines/#data>). Datasets should be deposited in an appropriate publicly available repository and details of the associated accession number, link or DOI to the datasets must be included in the Data Accessibility section of the article (<https://royalsociety.org/journals/ethics-policies/data-sharing-mining/>). Reference(s) to datasets should also be included in the reference list of the article with DOIs (where available).

Please submit a copy of your revised paper within three weeks. If we do not hear from you within this time your manuscript will be rejected. If you are unable to meet this deadline please let us know as soon as possible, as we may be able to grant a short extension.

And finally, all the best to you and your co-authors for the New Year; I hope you are all well given the current difficult circumstances, and here's hoping for a calmer 2021.

Best wishes,
 Loeske Kruuk
 Editor
 mailto: proceedingsb@royalsociety.org

Associate Editor Board Member

Comments to Author:

The authors have done a very thorough job with the revision, and have satisfied one of the original referees of their prior queries. I thank the authors for taking the time to consider all comments so carefully, and providing very thoughtful responses and revisions to the manuscript.

I also thank the authors for implementing Kenward Rogers F test approximations to estimate their parameter estimates for linear mixed models. Arising from this, I had a further query about the stats, pertaining to model specification of lmer models and estimation of parameter estimates for the interactions (I had discussed this potential issue in my original report on the original submission). Let's take the example on Line 256 of the revised manuscript, in which the authors model the interaction between mating status and evolutionary sex ratio on female fencing duration. The denominator degrees of freedom for the main effects in the model are around 6, and this seems very much correct (since there are 9 replicate populations). But the denominator degrees of freedom for the interaction between mating status and evolutionary sex ratio are much higher, and suggests misspecification of the unit of replication and resulting pseudoreplication. I was expecting to see denominator degrees of freedom of around 6 (given there are 9 replicate populations), but the value is close to 240 suggesting the error degrees of freedom represents the individual level rather than the population replicate level (this can be modified by inclusion of random slopes for the population replicate term). See Table 1 and Box 1 in Arnqvist et al 2020 Trends in Ecology and Evolution 35:329-335 for further clarification of the issue. This potential issue arises across all models in which the authors test for interactions, and I ask that they think about these models.

Editor (Loeske Kruuk) Comments to Author

I would just like to reiterate the AE's and the reviewer's assessment of this being an excellent paper, and that we appreciate the large amount of work that went into the data collection. As mentioned above, thanks for revising the statistical analyses. However, I agree with the AE's concern about the d.f.: for example, in the case of the mating status x sex ratio interaction, you are essentially testing the effect of mating status at the individual level, whereas the true number of independent replicates is fewer. The AE has recommended a straightforward solution to this, of including an interaction in the random structure between mating-status and population (in lmer, (1+ mating.status | replicatepopulation), also keeping mating.status as a fixed effect). Obviously this may change the interpretation of the significance of some of the interactions. I hope this is clear, and if not that you can discuss it with a statistician.

Given the need to clarify the presentation of some of the statistics, it would help the paper greatly if you could give full output of the models in a table, rather than just presenting test statistics in the text. This would allow you to show parameter estimates, SEs, and variance components for the different random effects, and would make the model structure much clearer - it took me a while to work it out.

Another suggestion for clarity would be to number the two experiments and refer to "Experiment I - Coevolved" and "Experiment II - Evolved": this would just make the structure clearer.

Reviewer(s)' Comments to Author:

Referee: 2

Comments to the Author(s).

The authors have done a good job of addressing my comments from the initial round of reviews, and the manuscript is now much clearer. I have a few small additional comments, listed below, that are not sufficient to warrant a full revision but could further improve the final manuscript. I look forward to seeing this paper in Proc B.

Lines 41-42 seem overly general, e.g., males may compete for territories, females may compete for reproductive opportunities, etc.

Lines 53-55 is there a specific idea here about what might co-vary with competition that is expected to produce increased aggression?

Line 116 just to be precise, there is evidence for aggression as a strategy for mate guarding in flies (<https://www.sciencedirect.com/science/article/abs/pii/S0003347215003206>). I don't think this change in aggression following mating would be relevant to your experimental design, however, since the males were separated from their mates during the aggression trials.

Author's Response to Decision Letter for (RSPB-2020-3053.R0)

See Appendix B.

Decision letter (RSPB-2020-3053.R1)

22-Feb-2021

Dear Dr Bath

I am pleased to inform you that your manuscript entitled "Sex ratio and the evolution of aggression in fruit flies" has been accepted for publication in Proceedings B.

The Associate Editor has noted some very minor corrections that can be corrected at proof stage, rather than you submitting a full new version of the manuscript (though if you would prefer to do the latter, please let us know immediately). You can expect to receive a proof of your article from our Production office in due course, please check your spam filter if you do not receive it. PLEASE NOTE: you will be given the exact page length of your paper which may be different from the estimation from Editorial and you may be asked to reduce your paper if it goes over the 10 page limit.

Open Access

Corresponding authors from member institutions (<http://royalsocietypublishing.org/site/librarians/allmembers.xhtml>) receive a 25% discount to these charges. For more information please visit <http://royalsocietypublishing.org/open-access>.

Paper charges

Sincerely,

Professor Loeske Kruuk

Associate Editor:

Comments to Author:

The authors have done an outstanding job with these final revisions, and I thank them. I note some minor typos at line 179 - 227 with spelling of word Experiment - consistently spelt EExperiment throughout the Statistical Analysis section of the Methods, and also other typo errors on line 181 (which probably come from an error made when using the find and replace function of Word). Perhaps also best to write in the random slope structure of the female behaviour models in the Statistical Analyses section of the methods. I think this will make a great paper and important contribution to the literature.

Appendix A

Dear Professor Kruuk,

Thank you for sending the helpful reviews of our manuscript and comments from the associate editor.

We have carefully revised the manuscript, and below we explain how we have addressed each of the comments. The most significant changes are, first, that we have followed the suggestion of replacing Wald Chi-squared tests with Wald F tests with Kenward-Roger degrees of freedom. This changed the statistical significance of our results on male low-intensity aggression and we have revised the manuscript accordingly. Second, we have revised our discussion of food patch occupancy following the reviewers' thoughtful comments, and added analysis and a figure to examine the relationship between aggression and food patch occupancy.

We believe that the revised manuscript is substantially improved, and hope that it is now suitable for publication in Proceedings B.

Thank you again for considering our manuscript.

With best wishes,

Eleanor Bath, Dani Edmunds, & Jen Perry

Associate Editor

1. This study is very interesting. The authors have evolved populations of *D. melanogaster* under different operational sex ratios (thus manipulating the intensity of sexual selection), and looked at corresponding changes in the frequency of male and female aggression. The paper was sent to two expert referees, each of whom enjoyed the study. Referee 1 found some of the description of the experimental design and behavioural measures ambiguous, and requiring of clarification. They have provided some helpful suggestions, as well as a series of detailed comments requiring attention. Referee 2 found it difficult to understand the statistical analyses described from line 199. I also struggle to follow what you have done, and why, and this needs to be explained more clearly. Referee 2 has also made other insightful comments, each requiring attention.

We are pleased that the reviewers enjoyed the study. We have addressed each of the reviewers' comments, including clarifying the description of our statistical analyses.

2. I also note that your linear mixed models have utilised Wald Chi-square tests, which are regarded as anti-conservative, and not ideal. Much better to apply Kenward Rogers or Satterthwaite F tests, which are less likely to give you Type 1 error; another massive benefit is you can then see the denominator degrees of

freedom to make sure that you have specified the random statement correctly [i.e. if your denominator dfs blow out beyond your number of replicates, you know you have a problem and inherent pseudoreplication (see Arnqvist, G. (2020). "Mixed Models Offer No Freedom from Degrees of Freedom." Trends in Ecology & Evolution.)]. Alternative is to use a parametric bootstrap. Surprisingly many models are inherently pseudoreplicated in ecology and evolution because the random intercept fails to account for the correct level of analysis for fixed interactions – this often requires fitting with random slopes.

Thank you for this helpful comment. We have followed this recommendation and replaced the Wald Chi-square tests with Wald F tests with Kenward-Roger degrees of freedom.

The statistical significance of most results remained the same. For our analysis of female aggression, some results for low-intensity aggression changing from marginally significant to non-significant (line 258). For our analysis of male aggression, our original finding of an effect of sex ratio on male fencing behaviour was no longer significant (line 233). We therefore no longer report an effect of sex ratio on the frequency or duration of male aggression, although effects on male aggression associated with food patch occupancy remain significant. We have edited the text accordingly.

We retain Wald Chi-squared tests for general linear models with binomial distribution because they are appropriate tests for these models.

Referee 1

1. This is a really interesting paper looking at how evolving under different sex ratios (and thus different levels of intrasexual competition) influences the evolution of male and female aggression behaviour in *Drosophila melanogaster*. The authors use well established experimental evolution lines and conduct a series of experiments/assays to get at this question from several angles and in so doing provide interesting and novel data on the topic of what drives the evolution of aggressive behaviour. For the most part the manuscript is clear and well written (although I do think it could be polished further for readability), the data collection and analysis methods are sound (although more detail and restructuring is needed in some areas) and the synthesis of literature and interpretation of results is balanced.

Most of the following detailed comments are relatively minor and should be easily addressed by the authors. They are either aimed at improving the clarity of the manuscript, driven by my interest in the study, or are simply typos that need correcting (sorry I can't help myself). The comments appear in the order they found in the manuscript

We are very pleased to hear the reviewer's positive view of our manuscript.

2. Line 26: add "s" to impact

We thank the reviewer for catching this. We have corrected the text (line 22).

1. Line 27: I think it is worth early on stating what you mean by competition or being more explicit by what type of competition you are referring to. Here I first thought you simply meant competition over food resources. In the abstract here it might be enough to say “intrasexual” competition, or the term you use in the intro “reproductive competition”.

We have changed the text to ‘intrasexual competition’ (line 24).

2. Line 34: replace “stronger” with “greater”

We have corrected the text (line 31).

3. Line 35: Which effect do you mean by “This effect”. Both the male and female responses to selection or just that of the female? Be explicit.

We now clarify that we are referring to the female response to selection (line 32).

4. Line 35-37: I find this sentence a little awkward. Also I’m not sure from your experiment whether you can invoke male stimulation of females via ejaculates as what drives coevolution. There could be other aspects of mating that drive this pattern. I guess previous evidence from drosophila might suggest this is the most likely pathway, but without that context in your abstract I would stick with what this study specifically can tell you, which is simply that the response is likely the result of coevolution.

We agree that other male traits could stimulate females, and have edited the abstract to delete the reference to ejaculates and to simplify the sentence. In the introduction, we speculate that the male ejaculate is a likely pathway based on our previous work (Bath et al. "Sperm and sex peptide stimulate aggression in female *Drosophila*", *Nature Ecology & Evolution* 1 (2017): 1-6) (line 32-34).

5. Line 37: where you say “our results” I think you are now talking about a different result. To make this clearer I would state the result explicitly or make it clearer you are now talking about something new by stating something like “we also found evidence for...”

We have clarified the text (line 35).

6. Line 70-80: This paragraph presents an interesting idea, but I found it really difficult at first to figure out what you are getting at. Are you basically saying that males could manipulate females to show increased aggression after mating because they benefit

from it (ie more eggs fertilised by them or eggs that are fertilised by them have more resources due to increased female access to food) without experiencing potential costs (ie injury)? That's cool, but I guess they only don't suffer the costs if the costs are delayed... and I wonder if it's more likely a side-effect eg males transfer something that increases female egg laying rate and this means they need greater access to food, thus have increased motivation to fight for it... Ok well the later part of this comment is just musings, but I do think the point of this paragraph could be made clearer.

We thank the reviewer and have edited the text for clarity (line 67-78). We agree that any immediate costs that females experience from aggression will be shared by a mating pair. Conflict between the sexes over optimal female aggression will arise when costs from aggression occur later. For example, costs might arise when aggression causes energy expenditure and damage that reduce later reproduction or lifespan. To the extent that a male does not sire a female's future offspring (e.g., because she has re-mated), he will not experience those costs.

We also agree that males might stimulate female egg laying, which might itself increase motivation to feed. We discuss this hypothesis later in the manuscript. Briefly, there is no evidence that increased female egg-production after mating directly causes increased feeding and aggression (Bath et al. "Sperm and sex peptide stimulate aggression in female *Drosophila*", *Nature Ecology & Evolution* 1 (2017): 1-6).

7. Line 85: preface competition with "intrasexual" or "reproductive" to avoid confusion

We have added 'intrasexual' (line 80).

8. Line 87: you say it later in the manuscript, but I think it is worth mentioning here, that males don't eat much as adults.

We thank the reviewer and have edited the text (line 85-86).

9. Line 91: The last sentence of this paragraph is awkward perhaps change to something like "Therefore differences in female aggression could represent a response to evolved differences in male ejaculates as well as the direct evolution of female behaviour."

We agree and have adjusted the wording (line 89-92).

10. Line 96-99: This sentence needs revising for flow and clarity

Revised the paragraph beginning on line 93.

11. Line 101: weird to have a sentence starting with “and” after a sentence starting with “finally”. Also this last sentence sounds tacked on and needs to be integrated better

We have deleted this sentence (see line 104).

12. Line 108-114: I found this description of the experiments really confusing. Rather than try and explain why I thought it was confusing. Here is a suggestion for how I think it could be made less confusing. Obviously I don't expect you to use my words, and I might have some of the details not quite right... but I think something like this would make the structure and purpose of your experiments clearer.

“We conducted two experiments in the first we measured intra-sexual aggression in virgin females, mated females and mated male that had evolved under male biased, equal and female biased evolutionary sex ratios. In this experiment all flies were mated with partners from the same replicate line. We tested both mated and virgin females because it is known that females show increased aggression after mating. However we only tested mated males, because they show no such change in behaviour. We then conducted a second experiment designed to look whether differences in female behaviour reflected direct evolutionary responses of aggression or reflected changes that resulted from the evolution of male traits. This second experiment involved mating females from our ESR lines to stock males as well as stock females to male from our ESR lines and measuring female aggression both before and after mating.”

We thank the reviewer for the suggestions and have adopted similar phrasing in our revision (see paragraph beginning line 110).

13. Line 133: What did you do with them after the trial? Put them back in individual containers?

We have edited the text to note that we discarded flies after trials (line 153).

14. Line 134: so virgin females were different individuals to mated females? ie they weren't the same females measured before and after mating? I guess that means virgins and mated females were tested at the same age? Could provide more detail and be more explicit around these points.

We have reworded this section to make this clearer. Virgin and mated females were different individuals and were tested at the same age (line 135-136).

15. Line 140: for females - were mated females always paired with mated females and virgin with virgin? I wonder if pairing with replicates could exacerbate differences between selection regimes due to escalation of contests??? Would it have been better to pair everyone with a standard competitor?

The reviewer is correct that we tested pairs of mated females and pairs of virgin females, and we have edited the text to make this clearer. We agree that an alternative experimental design is to use a standard competitor. We chose our design for two advantages it offers. First, contest behaviour can be strongly affected by the magnitude of difference in baseline aggression between rivals. Using a standard competitor type for both virgin and mated females across all treatment groups would mean that different groups of experimental females would be more or less similar to the standard competitor, and the degree of difference would change depending on the standard type selected. Our approach standardises the difference between competitors within a contest. Second, pairing individuals within selection lines means that all individuals face the type of competitor they will have encountered in their recent evolutionary history (please see also our response to the reviewer's comment 48) (see line 144-148 in text).

16. Line 147: delete "in"

Corrected

17. Line 150-165: Generally I think this paragraph could be much more succinct. More detail in following comments

We have edited the paragraph to be more succinct and in response to the following comments (see paragraph beginning line 155)

18. Line 157: I'm not sure how you sum the amount of time spent lunging since this is a count?

We have added text to clarify that we multiplied the number of lunges by the mean lunge duration. (Table S1)

19. Line 157: description of fencing behaviours could be incorporated into previous sentence using brackets

We have moved this text to the table describing aggressive behaviours (Table S1).

20. Line 160: The average duration of time spent on a food patch seems like a strange proxy for male territoriality to me. If both males spent a lot of time on the food patch.

We thank the reviewer for prompting us to think more carefully about behaviour related to territoriality. We have edited the text here and throughout the manuscript to refine and clarify our discussion of food patch occupancy (e.g. line 163-5, line 359-371). We have also added analyses and a figure to explore the relationship between aggression and food patch occupancy (line 203-209, figure 2). We found that the most aggressive individual within a dyad is able to occupy the food patch longer (line 239-241). We also found that the

relationship between aggression and food patchy occupancy differs with sex ratio: there was a weaker relationship in males from female-biased populations (line 241-245). We have added text to the discussion to set these analyses in context please see also our responses to comments 34, 40, 41 and 45).

21. Line 161: put info that males spend little time feeding in the intro.

We have added this to the intro (line 85-86).

22. Line 163-165: Why only in the coevolved experiment? what did you do in the other experiment? why only measure a subset of trials for low intensity? You have videos for all, and sample sizes aren't that much different, seems like a strange decision that should be justified.

We have edited the text to explain our approach. Behavioural data extraction is very time-consuming and some authors working on the co-evolved experiment had short-term contracts. We prioritized scoring female headbutts because headbutts are the main female aggressive behaviour (see caption in table S2)

23. Line 172-210: While your data analysis seems largely sound, the description is confusing and incomplete, this section needs a thorough revision. In so doing I also think it could be made more concise. Perhaps structuring it in a similar way to your results section would help? ie coevolved exp – males, females, female evolved experiment, male evolved experiment

We have edited the text to make this section clearer and more complete (see statistical analysis section starting line 176).

24. 172-177: is this just the "coevolved" experiment? what traits did you analyse?

We now clarify that we ran these models for all experiments (line 176).

25. Line 176: Why include observer for females and not males?

We have edited the text to note here also that a single observer extracted the male data, while two observers extracted the female data (line 185).

26. Line 177-179: Why not analyse these in the same model so you can look for an interaction? Also what traits were you looking at here?

We have edited the text to clarify the model and the traits (line 186-192). We required a different analysis for the evolved male experiment due to differences in the experimental design from the co-evolved experiment and the evolved female experiment. In the evolved male experiment, all females were standard females from stock populations, such that that we had a single virgin female group, along with three mated female groups (i.e., each mated to males from a different evolutionary sex ratio treatment). Therefore, we could not include an

interaction between mating status and sex ratio in the model for this experiment as we did with the other experiments.

27. Line 185-187: is this in addition to the Imm described above?

We have edited the text to emphasize that these were post-hoc tests performed after the linear mixed models (line 200-202).

28. Line 187: where do you describe what you did for the “evolved female” experiment

We have revised the text to make this clearer (please see our response to comment 24).

29. Line 188-191: Did you describe analysis of males above? I would put all male analyses together

We have edited the text for clarity and now group the male analyses together (line 176-181)

30. Line 194: add “s” to population

Corrected

31. Line 195: what determines whether including observer is applicable or not?

We have edited the text to emphasize that a single observer analysed the male data, while two observers analysed the female data (please see our response to comment 25).

32. Line 207-210: I wonder what this analysis adds to your paper – you don’t really talk about it later as far as I can tell. Perhaps you could put it in the supplement and just say above where you look at aggression that you did not include mating behaviour as a covariate, because it did not differ? Although, I must say I’m a little surprised it doesn’t differ, is this consistent with previous studies on these lines? If you think these analyses are important then you need to add more detail of model setup.

We agree that mating behaviour is not the focus of the paper and we have now condensed the text about mating behaviour to a note in the methods section to explain why mating behaviour did not need to be included as a covariate in models of aggression (line 192-193), and moved the results to the supplementary materials.

We had initially included this analysis because mating behaviour might influence aggressive behaviour (e.g., through differential transfer of male ejaculates, which influence female aggression; Bath et al. (2017) *Nature Ecology & Evolution* 1: 1-6), such that evolved differences in mating behaviour might underlie evolved differences in aggression. Our finding

that the evolutionary sex ratio did not affect mating latency or duration does not support this hypothesis.

Past studies offer mixed evidence for the evolution of mating latency and duration in response to experimental evolution under different sex ratios (see Table 5, Wensing et al. (2017) *Ecology and Evolution* 7, 10361-10378).

33. Line 212: throughout results be consistent in the number of decimal places (standard is 3) for reporting p values.

We now report all p values to 3 decimal places.

34. Line 223-224: This suggests to me that they are not really defending this resource as you suggest earlier. I wonder whether it is worth somehow taking into account the amount of space in the vial that it takes up to show that they actually defend it more than you would expect by chance? Or whether you need to think about your interpretation of the resource?

We have edited the text here and throughout to refine and clarify our discussion of territoriality (line 359-371), and added new analyses and a figure (figure 2, please see also our responses to the reviewer's comments 20, 40, 41 and 45). It is also the case that because male fights involve chasing, males move around while fighting, such that a fight might begin on a food patch and end elsewhere, and fights occurring adjacent to the food patch might relate to food patch defence but were scored as occurring off the food patch.

35. Line 250: do you mean it was "lower" in EQ females? or that it reduced after mating in these females?

We meant lower, but after adjusting models, we no longer find lower locomotion in EQ females.

36. Line 315: manipulated populations sex ratio... and then what?

We now state that we then assayed aggression after experimental evolution took place.

37. Line 317: "after evolving in a sex ratio skewed toward their own sex"

We have adopted this phrasing (line 334).

38. Line 317: here and elsewhere I find the phrase "evolve changes" weird. Consider rewording

We have changed the text to 'evolution' (line 334).

39. Line 322: “females” rather than individuals?

We believe that ‘individuals’ is more appropriate here because we are referring to both experimentally evolved females mating to stock males and experimentally evolved males mating to stock females (line 339).

40. Line 343: I guess I’m left wondering, if this is true, why don’t they spend more time defending it?

We have substantially revised our discussion of male behaviour in relation to food patches (please see our responses to comments 20, 34, 41 and 45).

41. Line 344: Could it be that decreased male aggression and territoriality is not directly a result of weaker male male competition, but that more females in this selection regime means that there is less female aggregation around the resource... ie some females are booted off, and so the food bottle becomes less valuable to defend. Would be nice to see some data on behaviour from the actual experimental lines if you have it... although not necessary for publication.

We agree that this is an interesting hypothesis. We now refer to the hypothesis and to data from our previous study of these experimentally-evolving populations, where we found that in fact female are more aggregated on food in female-biased populations (line 370-371) (please see also our responses to comments 20, 34, 40 and 45).

42. Line 346-361: This is an interesting argument but it a bit vague and I’m left wondering exactly what the predictions are you would expect for low and high intensity aggression for each of the ERS’s, and whether your results fit those predictions.

We have revised this argument and now put less emphasis on differences between high- and low-intensity aggression.

43. Line 366: delete “s” from findings

Corrected

44. Line 368: You don’t really discuss reasons differences between male and female responses in detail, but I wonder whether they could arise due to difference in the resource they are competing for?

We now elaborate on this possibility in the final section of the discussion (line 442-446).

45. Line 373-376: I'm not entirely sure that food patch occupancy actually reflects nutritional defence. If I remember correctly you averaged the time two females from the same line spent on the food patch so high values mean that both females spent lots of time on the patch, which would indicate they aren't defending it? But instead the opposite.

We have substantially revised our analysis and discussion of food-patch occupancy behaviour (please see also our responses to comments 20, 34, 40 and 41).

46. Line 389-390: What is the reasoning/evidence behind this statement?

We have edited the text to clarify the statement. We suggest that male ability to stimulate female aggression did not appear to evolve in the conditions of our experiment because experimentally-evolved males from different sex ratio treatments did not induce different levels of aggression in wild-type female mates (line 398-410).

47. Line 394: delete "s" from males

Corrected

48. Line 391-400: In your experiment you only mated pairs within their replicate. I wonder if you had performed crosses across lines but within selection regimes? You could have gotten a better handle on whether this result was due to coevolution per se or something to do with male and/or female traits?

We agree that the approach the reviewer suggests would be interesting, but we preferred the experimental design of this study for two reasons. First, we agree that population crosses seem like a design that should be useful for uncovering coevolution, but past studies have pointed out that it's difficult to distinguish hypotheses about coevolution from population crosses (Rowe et al. (2003) Detecting sexually antagonistic coevolution with population crosses, *Proceedings of the Royal Society B* 270: 2009-2016; Rowe & Day (2006) Detecting sexual conflict and sexually antagonistic coevolution, *Philosophical Transactions of the Royal Society* 361: 277-285). One challenge is that it is not clear whether male-female coevolution might arise through mutation-order effects specific to each line, or be driven in a similar direction in response to the sex ratio treatment (i.e., convergent coevolution). Furthermore, whatever the result from additional across-line crosses, we would still need to conduct the separate evolved-male and evolved-female experiments that we conducted in this study to determine whether differences in aggression were driven by evolution in male or females (or both).

Second, crossing across lines within selection treatments would greatly increase the sample size to a level unfeasible for our team (extracting the behavioural data is very time-intensive).

Figure 3: Would be worth mentioning in figure caption that data points reflect mean values for each replicate (or whatever it is they represent)

We now include a reference to the main text in the figure caption to direct the reader to a full interpretation of what the points represent.

Referee: 2

1. This paper explored the evolution of aggression in fruit flies by manipulating the sex ratio of populations for >75 generations. The authors tested the hypothesis that the overabundant sex should experience greater intra-sexual competition for resources and/or mates, and therefore evolve greater aggression. Interestingly, these hypotheses were supported only when flies mated with other members of their own population. This suggests that some kind of co-evolutionary process may have occurred, which is fascinating and perhaps unexpected.

The manuscript is in pretty good shape as it stands, and I think will make a nice contribution to Proc B.

We thank the reviewer for these positive comments.

2. I have a few concerns that I think the authors will be able to resolve fairly easily. From most important to least important, they are: I didn't understand the statistical approach used to distinguish between underlying inter-sexual genetic correlations and co-evolutionary processes, and this seems really important. Something about residuals...? Is this the equivalent of examining the inter-sexual correlation within populations, and if so, why not do that directly? This needs to be explained and justified more clearly. Also, a description of these 2 hypotheses (shared underlying genetics and co-evolution) isn't given until the methods section (around line 200), but it should be in the introduction (unless I missed it, if so sorry).

We have substantially re-worded our description of this statistical analysis for clarity, and introduce the hypotheses in the introduction (line 210-226). Our approach is analogous to estimating the inter-sexual correlation r_{MF} ; a precise estimate requires a quantitative genetic breeding design, which was beyond the scope of this study.

3. No evidence is provided that the effect of the stock population individuals would be generalizable across all "non-coevolved" individuals, but this is central to the interpretation of the results. i.e., using a different stock population could have

produced different results. I think the authors need to add this as a caveat to the discussion section

We have added text to describe our choice of stock individuals in our methods section (line 121-123). The stock population was of the same genetic background (Dahomey) as the experimentally-evolved populations, so we believe it is the most appropriate genetic background for stock individual. It is a good point that interactions between male and female genes might have generated a different result with a different stock population. However, previous work has demonstrated that male stimulation of female aggression is consistent in magnitude when different stock populations are crossed (Bath et al., "Temporal and genetic variation in female aggression after mating." *PLoS one* 15 (2020): e0229633).

4. More needs to be explained about the ecological conditions in which selection occurred. This wasn't mentioned until the discussion, but I was wondering about it when reading the results for patch occupancy.

We thank the reviewer for this . We have text to describe the ecological conditions during experimental evolution (see supplementary methods), including discussion of food patch occupancy patterns observed by Rostant et al (2020; *Evolution Letters* 4: 54-64) in a separate study conducted on these populations (line 370-371).

5. Acronyms make the paper much harder to read and are not necessary when they just stand for 2 words. Each time I saw "FB" and the others I had to pause and try to remember what it means. Please switch these to real words.

Corrected

Appendix B

8th February 2021

Dear Prof. Kruuk,

Thanks very much for your positive comments about our manuscript and for the very helpful feedback from yourself, the associate editor and the reviewer.

We greatly appreciate the comments about random effects in our models and for pointing us to the useful reference. This has allowed us to correct the models in this and other manuscripts we're preparing on these experimentally-evolving populations. Overall, our results remain similar, but we now find no significant interaction between mating status and evolutionary sex ratio for two female behaviours (fencing and food patch occupancy). However, the interaction remains significant for high-intensity aggressive behaviour (head-butting).

We now give the full output from each model in the supplemental materials.

Please find our responses to the comments below. Line numbers refer to lines in the new manuscript version with all the changes accepted.

Thank you again for considering our manuscript for Proceedings B.

With best wishes,
Eleanor Bath
On behalf of all the authors

Associate Editor Board Member

1. The authors have done a very thorough job with the revision, and have satisfied one of the original referees of their prior queries. I thank the authors for taking the time to consider all comments so carefully, and providing very thoughtful responses and revisions to the manuscript.

We thank the associate editor for these comments.

2. I also thank the authors for implementing Kenward Rogers F test approximations to estimate their parameter estimates for linear mixed models. Arising from this, I had a further query about the stats, pertaining to model specification of lmer models and estimation of parameter estimates for the interactions (I had discussed this potential issue in my original report on the original submission).

Let's take the example on Line 256 of the revised manuscript, in which the authors model the interaction between mating status and evolutionary sex ratio on female fencing duration. The denominator degrees of freedom for the main effects in the model are around 6, and this seems very much correct (since there are 9 replicate populations). But the denominator degrees of freedom for the interaction between mating status and evolutionary sex ratio are much higher, and suggests misspecification of the unit of replication and resulting pseudoreplication. I was expecting to see denominator degrees of freedom of around 6 (given there are 9 replicate populations), but the value is close to 240 suggesting the error degrees of freedom represents the individual level rather than the population replicate level (this can be modified by inclusion of random slopes for the population replicate term). See Table 1 and Box 1 in Arnqvist et al 2020 Trends in Ecology and Evolution 35:329-335 for further clarification of the issue. This potential issue arises across all models in which the authors test for interactions, and I ask that they think about these models.

We are very grateful to the AE for pointing out this important issue and for the very helpful example and reference. We have corrected the models and the degrees of freedom now reflect the number of experimentally evolving populations. Overall, the results remain similar, but two results have changed: we no longer find a significant interaction between mating status and sex ratio for female fencing or for female food patch occupancy. The interaction remained significant for female head-butting.

Editor (Loeske Kruuk)

1. I would just like to reiterate the AE's and the reviewer's assessment of this being an excellent paper, and that we appreciate the large amount of work that went into the data collection.

Thank you, we are so pleased to hear this!

2. As mentioned above, thanks for revising the statistical analyses. However, I agree with the AE's concern about the d.f.: for example, in the case of the mating status x sex ratio interaction, you are essentially testing the effect of mating status at the individual level, whereas the true number of independent replicates is fewer. The AE has recommended a straightforward solution to this, of including an interaction in the random structure between mating-status and population (in lmer, (1+mating.status|replicatepopulation), also keeping mating.status as a fixed effect). Obviously this may change the interpretation of the significance of some of the interactions. I hope this is clear, and if not that you can discuss it with a statistician.

Thank you for this comment and the very useful R script, which we have implemented in correcting the models. As described above, two interactions that were previously statistically significant are now not significant, but the overall picture of our results remains the same.

3. Given the need to clarify the presentation of some of the statistics, it would help the paper greatly if you could give full output of the models in a table, rather than just presenting test statistics in the text. This would allow you to show parameter estimates, SEs, and variance components for the different random effects, and would make the model structure much clearer - it took me a while to work it out.

We now include the full model output for all models in the supplementary materials (we found that we could not keep the manuscript under the size limit if we included another table in the text).

4. Another suggestion for clarity would be to number the two experiments and refer to "Experiment I - Coevolved" and "Experiment II - Evolved": this would just make the structure clearer.

Thank you, we have done this.

Reviewer 2

1. The authors have done a good job of addressing my comments from the initial round of reviews, and the manuscript is now much clearer. I have a few small additional comments, listed below, that are not sufficient to warrant a full revision but could further improve the final manuscript. I look forward to seeing this paper in Proc B.

We thank the reviewer for these positive comments.

2. Lines 41-42 seem overly general, e.g., males may compete for territories, females may compete for reproductive opportunities, etc.

We thank the reviewer and have edited the text to emphasize that these patterns are not universal (Line 42).

3. Lines 53-55 is there a specific idea here about what might co-vary with competition that is expected to produce increased aggression?

We have added text to suggest co-varying factors (conspecific density and resource distribution (Line 55)).

4. Line 116 just to be precise, there is evidence for aggression as a strategy for mate guarding in flies (<https://www.sciencedirect.com/science/article/abs/pii/S0003347215003206>). I don't think this change in aggression following mating would be relevant to your experimental design, however, since the males were separated from their mates during the aggression trials.

We thank the reviewer and have added text to acknowledge mate guarding, including the citation (Line 115).